# Measurement of Residual Stress and Young’s Modulus on Micromachined Monocrystalline 3C-SiC Layers Grown on <111> and <100> Silicon

**DOI:** 10.3390/mi12091072

**Published:** 2021-09-03

**Authors:** Sergio Sapienza, Matteo Ferri, Luca Belsito, Diego Marini, Marcin Zielinski, Francesco La Via, Alberto Roncaglia

**Affiliations:** 1Institute for Microelectronics and Microsystems (IMM), CNR, 40129 Bologna, Italy; sapienza@bo.imm.cnr.it (S.S.); belsito@bo.imm.cnr.it (L.B.); marini@bo.imm.cnr.it (D.M.); roncaglia@bo.imm.cnr.it (A.R.); 2NOVASIC, 73190 Saint-Baldoph, France; mzielinski@novasic.com; 3Institute for Microelectronics and Microsystems (IMM), CNR, 95121 Catania, Italy; francesco.lavia@imm.cnr.it

**Keywords:** 3C-SiC, MEMS, Young’s modulus

## Abstract

3C-SiC is an emerging material for MEMS systems thanks to its outstanding mechanical properties (high Young’s modulus and low density) that allow the device to be operated for a given geometry at higher frequency. The mechanical properties of this material depend strongly on the material quality, the defect density, and the stress. For this reason, the use of SiC in Si-based microelectromechanical system (MEMS) fabrication techniques has been very limited. In this work, the complete characterization of Young’s modulus and residual stress of monocrystalline 3C-SiC layers with different doping types grown on <100> and <111> oriented silicon substrates is reported, using a combination of resonance frequency of double clamped beams and strain gauge. In this way, both the residual stress and the residual strain can be measured independently, and Young’s modulus can be obtained by Hooke’s law. From these measurements, it has been observed that Young’s modulus depends on the thickness of the layer, the orientation, the doping, and the stress. Very good values of Young’s modulus were obtained in this work, even for very thin layers (thinner than 1 μm), and this can give the opportunity to realize very sensitive strain sensors.

## 1. Introduction

Young’s modulus and residual stress are key properties for structural microelectromechanical systems (MEMS) layers. Among the unique material properties of SiC, the high value of Young’s modulus and the relatively low mass density permit SiC to achieve higher resonant frequencies compared to other materials. SiC, given its larger E/ρ (where E is the Young’s modulus and ρ is the material density), yields devices that, for a given geometry, operate at significantly higher frequencies than are otherwise possible using conventional materials [1,2].

The heteroepitaxy of SiC on Si substrates results in the hetero-structure 3C-SiC/Si, which is a very interesting material system for micro- and nano-electromechanical systems. Unfortunately, the growth of 3C-SiC on Si is affected by the high mismatch in the lattice parameters (about 20%) and the thermal expansion coefficients (about 8%) between the two materials. The large mismatch is blamed for the generation of a high number of defects, such as misfit dislocations, twins, and stacking-faults (SFs) at the interface. [3,4,5,6]. The defect inside the film alters the crystal structure of the system and can modify the elastic properties of the materials. For this reason, the use of SiC in Si-based MEMS fabrication techniques has been very limited. 

A common observation in the 3C-SiC/Si system is that extended defects created at the 3C-SiC/Si interface (SFs, µ-twins) are mutually annihilating during growth, and their density reduces with increasing distance from the interface. The reduction of defect density is reflected by a decrease of the full width at half maximum (FWHM) of the rocking-curves of 3C-SiC X-ray diffraction peaks, while the thickness of the layer increases. Another confirmation of the improvement of 3C-SiC quality with increasing thickness is the decrease of the FWHM of the Raman 3C-SiC transverse optic (TO) peak.

In previous papers [7,8] it has been observed that the Young’s modulus of 3C-SiC film was increasing with increasing thickness of the 3C-SiC epilayer. It was quite straightforward to associate this effect with the progressive reduction of density of extended defects and improvement of the structural quality of the epilayer.

Residual stress strongly depends on the orientation of the underlying Si substrate [9]. It is also influenced by the growth conditions (during carbonization and the CVD step). An influence of the doping of the layer on the stress was also reported [10,11]. Finally, the distribution of defects inside the layer [12,13] plays a role.

Recent results reported on 3C-SiC on Si MEMS resonators have put into evidence the relation between the presence of a high tensile stress on the material and the possibility to obtain ultra-high Q-factor resonators with it [14]. Since Q-factor is closely related to the resolution of resonant sensors [15,16,17], the possibility to obtain SiC layers with high Young’s modulus and controlled tensile stress is attractive, because it may enable the fabrication of high-resolution sensors using MEMS resonators.

The complete characterization of the Young’s modulus and residual stress of monocrystalline 3C-SiC layers with different doping types grown on <100> and <111> oriented silicon substrates is reported in this paper. Moreover, test structures with different designs are micromachined on the substrates to evaluate the achievable yield on the 3C-SiC layers in MEMS fabrication.

## 2. Materials and Methods

### 2.1. 3C-SiC Growth

3C-SiC thin films used for this study were deposited in a previously described horizontal, low pressure, resistively heated hot wall chemical vapor deposition (CVD) system with a rotating sample holder [18]. Hetero-epitaxial 3C-SiC growth on Si substrates was achieved using a classical two-step process with a purified hydrogen (H_2_)/argon (Ar) mix as carrier gas, and silane (SiH_4_) and propane (C_3_·H_8_) as Si- and C-precursors. N-type doping with nitrogen was obtained by introducing N_2_ to the reactor chamber. For p-type doping with aluminum, we used tri-methyl-aluminum diluted in H_2_ (TMA). Fine-tuning of Al incorporation was performed by adding HCl to the process. Three types of 100 mm substrates were used: 525 µm thick on-axis (100) bare Si, 1000 µm thick on-axis (111) bare Si, and (100) SOI (10 µm device layer + 0.5 µm buried oxide).

In order to explore in detail the evolution of elastic/mechanical properties of 3C-SiC material, epilayers with different orientations ((100) and (111)), thicknesses (0.4 µm–1.3 µm), doping type (n-type, p-type), and doping level (from NID—non intentionally doped, <1 × 10^16^ cm^−3^, to ~5 × 10^19^ cm^−3^) were prepared. The list of samples processed for the needs of this study is given in Table 1.

Current characterization of the epilayers included optical microscopy imaging, thickness cartography from Fourier-transform Infrared (FTIR) spectrometry, optical determination of epiwafer deformation, and contactless sheet resistance measurement (eddy current approach). Atomic force microscope (AFM) imaging and X-ray diffraction (XRD) rocking curve measurements were performed on selected samples.

### 2.2. Fabrication

The process flow adopted to fabricate the micromachined test structures on bulk and SOI wafers is schematically represented in Figure 1, referring to the case of the SOI wafer.

The fabrication on the bulk wafers was identical. The only difference was the fact that while the fabrication flow in Figure 1 is a surface micromachining process exploiting the SOI device layer as a sacrificial layer, on bulk wafer it became a front-side bulk micromachining process in which the gap under the released structures at step 6 was determined by the duration of the SF_6_ release etching.

To etch the SiC layer grown on silicon, a mask composed of two stacked SiO_2_ (1.5 µm thick) and polycrystalline silicon (300 nm thick) layers was used. The composite mask was patterned by lithography and self-aligned RIE etching of polysilicon and SiO_2_ in sequence (step 4 in the figure). Next, the pattern was transferred onto 3C-SiC by another RIE process (step 5), up to the silicon surface. During the SiC etching, the thin polysilicon layer used in the hardmask was completely removed, whereas some residual SiO_2_ remained, as shown in the figure. Such a residual SiO_2_ mask was exploited to protect the SiC layer during the release step performed by isotropic silicon plasma etching (step 6). Afterwards, the remaining SiO_2_ was removed by HF vapor etching (step 7). An additional TMAH Si etching step, introduced to increase the under-etching of the structures without damaging SiC, concluded the process (step 8).

The following process parameters were adopted in the fabrication: polysilicon RIE at step 4 was executed by capacitively coupled (CC) plasma based on SiCl_4_ (150 W RF power, 36 mTorr background pressure, 16 sccm SiCl_4_ gas flow); SiO_2_ RIE at step 4 by CC plasma based on CHF_3_ (150 W, 38.5 mTorr, 25 sccm flow); 3C-SiC RIE at step 5 by CC plasma based on CHF_3_ and O_2_ (150 W, 100 mTorr, 25 sccm CHF_3_ and 12.5 sccm O_2_ flows); Si isotropic etching at step 6 by CC plasma based on SF_6_ (80 W, 75 mTorr, 64 sccm); SiO_2_ etching at step 7 was carried out by HF vapor holding the substrate at 37 °C; and Si etching at step 8 by 5 wt.% tetra-methyl-ammonium hydroxide (TMAH) aqueous solution with added 0.5 wt.% ammonium peroxidisulphate. SiO_2_ deposition by low-pressure chemical vapor deposition (LPCVD) at step 3 was performed from SiH_4_ and O_2_ at 180 mTorr and 420 °C; undoped polysilicon at step 3 was deposited from SiH_4_ at 160 mTorr and 595 °C. Contact lithography at step 4 was executed by Fujifilm UV6 0.6 µm thick photoresist. 

The mask used in the fabrication was composed of the replication of a 7 × 7 mm^2^ die containing a variety of micromachined test structures, including strain gauges, double and single-clamped beam arrays with different lengths, and double-ended tuning fork (DETF) resonators (Figure 2).

### 2.3. Characterization

A possible method that can be used to estimate the residual stress on a deposited layer is the measurement of the resonance frequency of a released double clamped beam fabricated with it. Such resonance frequency f_r,i_ can be modelled with the following formula [19]:(1)$$\;\;\;\;\;\;\;\;\;\;\;\;\;\;\;\;\;\;\;\;\;\;\;\;\;f_{r,i}=\frac{i^{2}\pi }{2L^{2}}\sqrt{\frac{\mathrm{EI}}{\rho A}}\sqrt{1+\frac{{\mathrm{SL}}^{2}}{i^{2}{\mathrm{EI}\pi }^{2}}}\;\;\;\;\;\;\;\;\;\;\;\;\;\;\;\;\;\;\;\;\;\;\;\;\;$$
where E is the Young’s modulus of the material, S is the tensile force applied to the beam (resulting from the residual stress), i is an integer mode index (the order of the resonance frequency), L is the beam length, ρ is the density of the material, and I and A are the cross sectional moment of inertia and area of the beam, respectively. 

Under high stress conditions, Equation (1) can be simplified as [20]
(2)$$\;\;\;\;\;\;\;\;\;\;\;\;\;\;\;\;\;\;\;\;\;\;\;\;\;f_{r,i}=\frac{i^{2}}{2L}\sqrt{\frac{\sigma_{0}}{\rho }}\;\;\;\;\;\;\;\;\;\;\;\;\;\;\;\;\;\;\;\;\;\;\;\;$$
where σ_0_ = S/A is the residual stress of the film. Apart from the residual stress σ_0_, Equation (2) only depends on the length of the beam (L), the density of the material (ρ) and the resonance order (i). To determine the residual stress, we used Equation (2) to fit the first-order resonance frequencies of double-clamped beams with different lengths, fabricated with the process flow described in Section 2.2. In the fitting procedure, we used σ_0_ as a fitting parameter, assuming a material density ρ of 3210 kg/m^3^ and using the nominal value of the beam length L. In all our films, the residual stress was tensile and sufficiently high to make Equation (2) an excellent approximation of Equation (1).

For the measurement of the resonance frequencies of the double clamped beams, we used a typical experimental setup [21,22,23], as shown in Figure 3. 

In the setup, the output port of a network analyzer (Anritsu MS2036C) was used to control the vibration of a piezoelectric disc (PRYY+0189) from PI, using the proper amplification stage to reach an AC voltage of 4 V_pp_. Exploiting the capabilities of the network analyzer, the frequency of the AC voltage was varied across a range centered on the expected resonance frequency of the beam under measurement, which was contained in a micromachined sample attached to the piezoelectric disk. The vibration of the beam induced by the piezoelectric actuation was detected by a laser doppler vibrometer (OFV-534) that produced a signal proportional to the vibration velocity of the microstructure through the OFV-5000 controller. Such signal was fed into the input port of the network analyzer, enabling the measurement of the vibration velocity of the beam close to its resonance. In this way, focusing the laser source on the beam, the transfer function displayed by the network analyzer showed a peak at the resonance frequency of the microstructure, which could be consequently measured accurately. The piezoelectric disc and the sample were both kept inside a vacuum chamber equipped with a transparent window during the laser measurements, maintaining an internal pressure of 0.075 mBar to reduce the air damping effect.

After measuring the residual stress as described above, the residual strain of the film was also determined independently, using the classical released strain gauge shown in Figure 4. As proposed in [24], we determined the residual strain of the SiC film by measuring the lateral deflection of the released strain gauge at the optical microscope and matching the obtained value with a finite-element simulation of the gauge deflection, in which the residual strain was used as a fitting parameter.

The geometrical parameters of the strain gauge are reported in Table 2.

In order to determine the residual strain, the displacement of the horizontal bar was first measured by optical observation using a Nikon microscope, and then the same shift was replicated via simulations using COMSOL Multiphysics. The simulation was configured with the same geometrical dimensions and material of the strain gauge, and then the residual strain was estimated with a numerical iterative method in order to obtain the same deflection determined experimentally. 

The Young’s modulus E was calculated after determining the residual strain (ε_0_) and stress (σ_0_) using Hooke’s law:(3)$${\;\sigma }_{0}={E\varepsilon }_{0}$$

Figure 5 shows an example of the shift of the strain gauge simulated with COMSOL Multiphysics.

## 3. Results

### 3.1. 3C-SiC Growth

The films prepared for this study followed typical characteristics of as-grown 3C-SiC/Si material presented in previous papers (see details in [25] and example in Figure 6). 3C-SiC thickness dispersion was below 5%. The RMS roughness (5 × 5 µm^2^) was in the 5–10 nm range for (100) epilayers and the 2–5 nm range for (111) orientation. The epiwafer deformation (bow) of NID samples was below 15 µm for 3C-SiC/Si(111) and below 10 µm for 3C-SiC/Si(100)—these are standard values for chosen substrate/film thicknesses. For (100) oriented samples with strong Al doping, a concave bow increasing with Al incorporation was observed, as previously reported in [26].

### 3.2. Fabrication

The yield of the fabrication was generally good on all the wafers, even for the layers with the highest residual stress. Moreover, the process flow described earlier allowed released structures with submicrometric size to be obtained, both on gaps and beams, as was observed on the test DETF geometries designed on the mask.

In Figure 7, an example of microscope inspection employed to evaluate the fabrication yield is reported. The device shown in the picture is a DETF resonator, in which the lateral electrodes reported in the magnification have a coupling gap around 0.65 µm and a tine width of 0.79 µm. The measurement of the submicrometric features of the fabricated structures was performed using the measuring tool of a Nikon optical microscope at a magnification of 100✕. The microscope inspection was an effective method to evaluate the fabrication yield in general, and in particular of the DETF geometries, in which the most critical features were designed (gaps and tine widths scaled down to 0.6 µm). 

Through the optical inspection, the correct release from the substrate of the MEMS structures could also be evaluated straightforwardly, because, as can be seen in Figure 7, the color of the SiC released parts was different than that of the SiC film still anchored to the silicon substrate.

Although the overall yield of the MEMS fabrication process was good, even on the most critical designed geometries, a clear difference was observed between the <100> and <111> substrates. On <100> substrates, the yield was close to 100% independently of the geometry of the released structure. On <111> substrates, instead, the yield was lower for two reasons. First, the residual stress was so highly tensile on the SiC layers grown on these substrates that it was able to exceed the yield strength on the material on peculiar structures, in which stress concentration phenomena occurred after release. On such structures, the yield was lower than 100%.

The second reason for the lower yield obtained on the <111> substrates was the appearance of cracks on the layers after SiC RIE etching (Figure 8). Such cracks were very long and propagated both in the SiC layer and in the Si substrate underneath, breaking all the structures that they came across. The problem of cracks was particularly severe on the <111> substrates on which the highest values of residual stress were measured (wafers W1, W5, and W6 in Table 1).

### 3.3. Characterization

We performed residual stress measurements using the method described in Section 2.3 exploiting arrays of double clamped beams, like those shown in Figure 9, as test structures. The beams had a fixed width of 16 μm, thickness dependent on the measured SiC layer (please see Table 1), and length varying from 200 μm to 1000 μm (ΔL = 200 μm). 

The normalized amplitude spectrum measured with the experimental setup shown in Figure 3 on a double clamped beam appeared as shown in Figure 10, in which an example of the optical characterization of a double-clamped beam from wafer W5, characterized by a length L of 800 μm, is reported. As can be seen from the plot, the device characteristic shows a clear resonance peak corresponding to its absolute maximum, from which a resonance frequency around 372 kHz can be determined. 

Figure 11a,b show the measured first-order resonance frequencies versus the length of the beam for all the samples with n doping grown on <111> and <100> silicon substrates. Figure 12 shows the same plots for the samples with p doping grown on <100> silicon substrates. The plots also report the fitting of the experimental data performed with Equation (2), assuming a material density of 3210 kg/m^3^, which was used to estimate the residual stress of the layers. 

As can be observed from the plots, the data fitting of the experimental results was found to be in very good agreement with Equation (2), with a least-squares $R^{2}$ coefficient around 0.98–0.99 for each sample. Subsequently, as described in Section 2.3, we measured the residual strain using strain gauges fabricated close to the beam array.

An example of microscope observation at 50x magnification of a strain gauge (sample P2) is shown in Figure 13. From the image, the horizontal shift of the indicating bar was measured, and the strain determined by fitting the deflection of the gauge with an FEM simulation, as explained in Section 2.3.

Figure 14 shows the simulation of the displacement for sample P2 with a Young’s modulus of 379 GPa and residual stress of 360 MPa. In Table 3, a complete overview of the measurement results for all the samples investigated is summarized. In particular, the table reports in the three rightmost columns the residual stress determined by the fitting of the experimental resonance frequency data acquired on the double clamped beam arrays using Equation (2) (Figure 11 and Figure 12), the residual strain determined by observing the shift of the released gauges with the aid of the finite-element simulations, and the Young’s modulus evaluated by calculating the ratio between the two former parameters, using Equation (3).

## 4. Discussion

The method that we have adopted for the estimation of Young’s modulus consists of determining the residual strain and stress with independent measurements and deriving E by calculating the ratio of the two parameters. This is conceptually simple and was also mentioned as a possible application of the first MEMS strain gauges presented in the literature by their inventors [27]. Nevertheless, to the best of our knowledge, it was never applied in practice to simultaneously determine Young’s modulus and residual stress/strain on SiC or other types of MEMS films. Consequently, a direct comparison of our results with measurements performed on similar films and with the same technique is not possible.

However, there are measurement results of 3C-SiC on Si Young’s modulus obtained with other methods that have been reported in the literature. One of the most frequently used method for Young’s modulus relies on the measurement of the mechanical resonance frequency of cantilevers, to which E is correlated through a formula also depending on the geometrical dimensions of the microstructure (length, width, and thickness) and on the density of the material.

Such a method was analyzed in detail in [28], in which cantilevers of different lengths were utilized to evaluate Young’s modulus of SiC on <111> Si layers. The authors found that, using the simple Bernoulli formula for the resonance frequency of the cantilevers, a satisfactory fitting of all the measured lengths could not be obtained. This happened because of the effect of stress gradient and of the consequent out of plane bending of the cantilevers on the resonance frequency, for which a correction of the Bernoulli formula was necessary. With such correction, the authors were able to fit all the cantilever lengths with a single value of E, obtaining a correct measurement.

With our method, the presence of a stress gradient was not problematic, since the planarity of the double-clamped beams that we used were not affected at all by the stress gradient, because of their mechanical boundary conditions. Using Formula (2), we were able to fit well all the measured lengths using a single value of the residual stress. In addition, the strain gauges that we used were similar to a double clamped beam and were relatively unaffected by the stress gradient.

In Figure 15 and Figure 16, we compared our results with other data reported in previous papers [29,30,31,32,33,34,35].

As can be seen from the comparison shown in the figures, our results are in reasonable agreement with other data reported in the literature on similar SiC on Si samples, following the well know trend that relates Young’s modulus with the thickness of the grown material. On the samples grown on <111> silicon, the presence of n-type doping seems to lead to an increase of Young’s modulus compared to the trend observed in NID samples with comparable thickness. On the contrary, p-type doping on <100> grown samples apparently causes a decrease of E compared to NID samples with similar thicknesses.

Concerning residual stress and strain, it is quite clear that the growth on <111> silicon permits very high values of residual stress to be obtained, two to three times higher than those achieved by growing on <100>. Moreover, the presence of N-type doping seems to increase the residual stress on <111> grown samples and decrease it on <100> grown samples. The opposite occurs with p-type doping on <100> grown samples, on which the residual stress is increased compared to the NID case.

## 5. Conclusions

In this paper, a method to estimate Young’s modulus with independent measurements of residual stress and strain has been applied for the first time to hetero-epitaxial 3C-SiC grown on silicon. Residual stress was estimated by measuring the resonance frequency of double clamped beams with different lengths, while residual strain was evaluated by measuring the deflection of micromachined strain gauges, with the aid of finite-element simulations. Young’s modulus was estimated according to Hooke’s law by calculating the ratio between the independently measured values of residual stress and strain. The results obtained were consistent with other data previously reported in the literature, showing the typical dependence of Young’s modulus on the thickness of the grown material. Since both n-type and p-type SiC layers were investigated, the reported experiments also allowed us to observe the effect of doping, which seemed to lead to an enhancement of Young’s modulus on n-type SiC grown on <111> silicon and to a decrease of it on p-type SiC grown on <100> silicon. Considering tensile residual stress, which was present on all the layers, extremely high values were obtained on relatively thin, n-type SiC layers grown on <111> silicon, even though a decrease of the MEMS fabrication yield was observed on these samples due to the appearance of cracks after micromachining. Some enhancement of the tensile stress was also observed introducing p-type doping in <100> grown layers. We believe that these results may be helpful in choosing the best growth conditions to achieve thin 3C-SiC films on silicon with high tensile stress and Young’s modulus, which, considering recent data reported in the literature, should be ideal for the fabrication of sensors based on ultra-high Q-factor vertical resonators.

## Figures and Tables

**Figure 1 micromachines-12-01072-f001:**
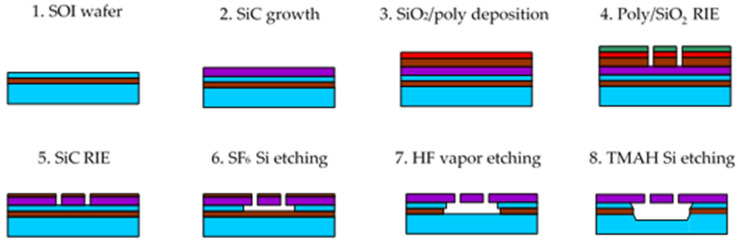
Process flow for the fabrication of the 3C-SiC test structures on SOI wafer.

**Figure 2 micromachines-12-01072-f002:**
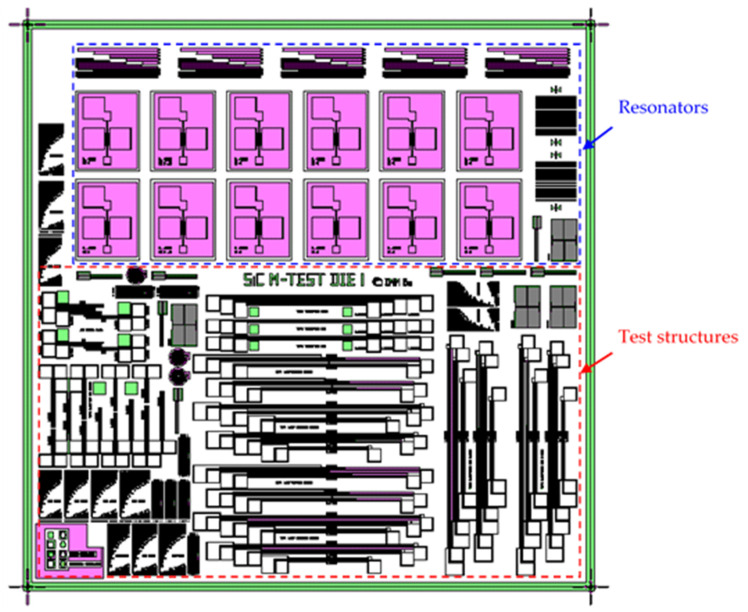
Micromachined test structures mask layout.

**Figure 3 micromachines-12-01072-f003:**
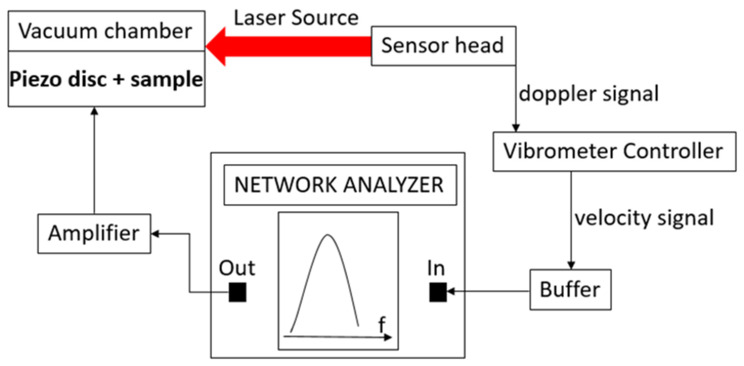
Schematic of the experimental setup used to measure the resonance frequency of the double clamped beams.

**Figure 4 micromachines-12-01072-f004:**
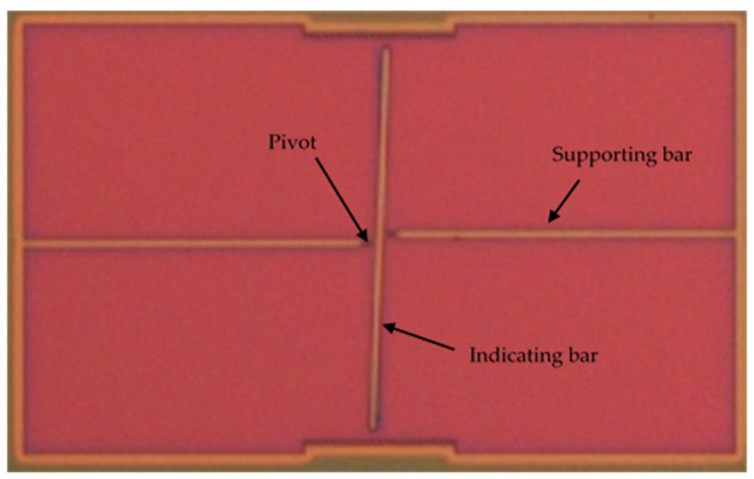
Picture of released strain gauge taken at the optical microscope.

**Figure 5 micromachines-12-01072-f005:**
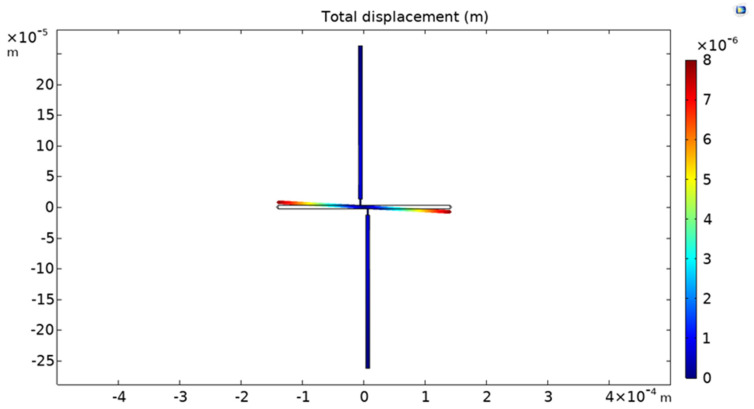
Strain gauge displacement simulated with COMSOL Multiphysics.

**Figure 6 micromachines-12-01072-f006:**
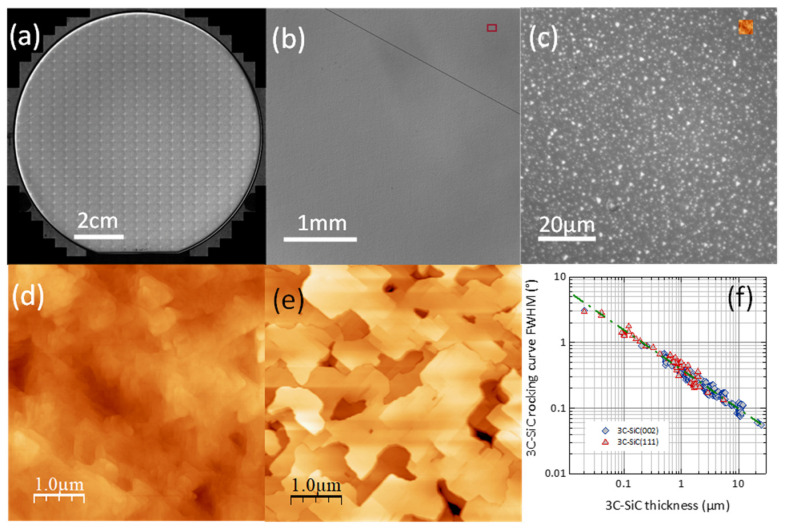
(**a**) Composite image of the 3C-SiC/Si(111) epiwafer W1, (**b**) broad field (×2.5 objective) image showing high quality as-grown surface with locally appearing cracks, (**c**) surface detail (x100 objective) with visible interfacial voids, (**d**) AFM image of the 3C-SiC(111) surface, (**e**) AFM image of the 3C-SiC(100) surface, (**f**) reduction of 3C-SiC rocking curve FWHM with increasing film thickness, attesting progressive reduction of intrinsic defect density.

**Figure 7 micromachines-12-01072-f007:**
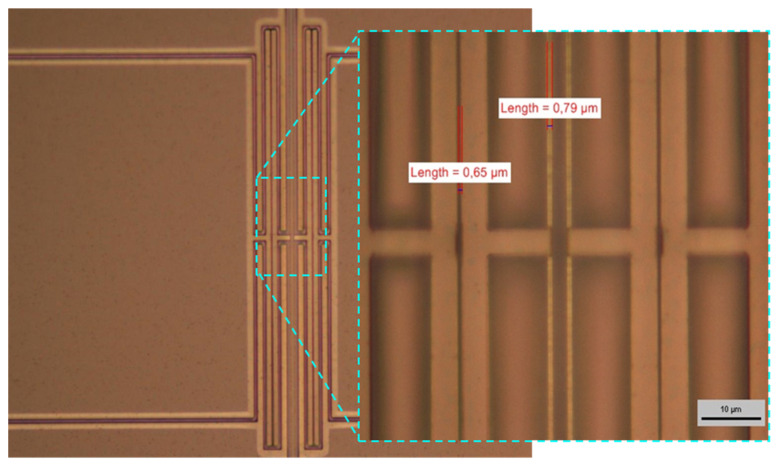
Submicrometric features on released DETF structures fabricated in 3C-SiC grown on <100> silicon.

**Figure 8 micromachines-12-01072-f008:**
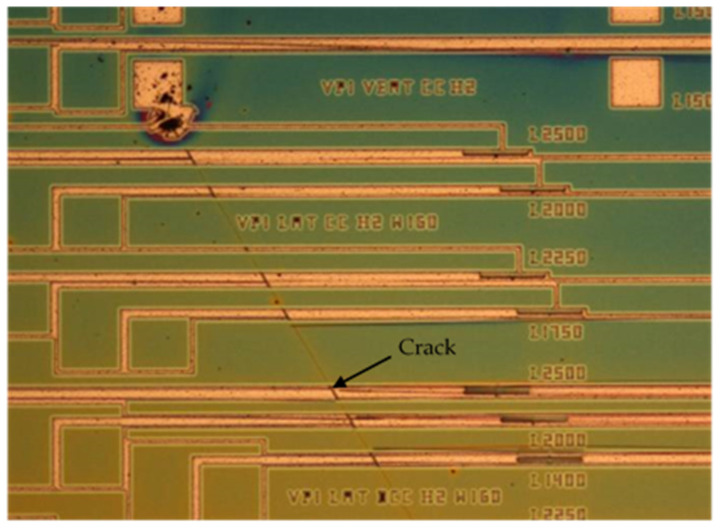
Cracks appearing after SiC micromachining on <111> substrates.

**Figure 9 micromachines-12-01072-f009:**
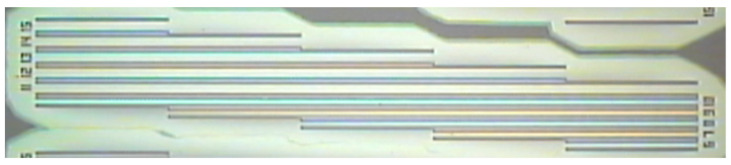
Array of double clamped beams with length variable from 200 μm to 1000 μm (ΔL = 200 μm).

**Figure 10 micromachines-12-01072-f010:**
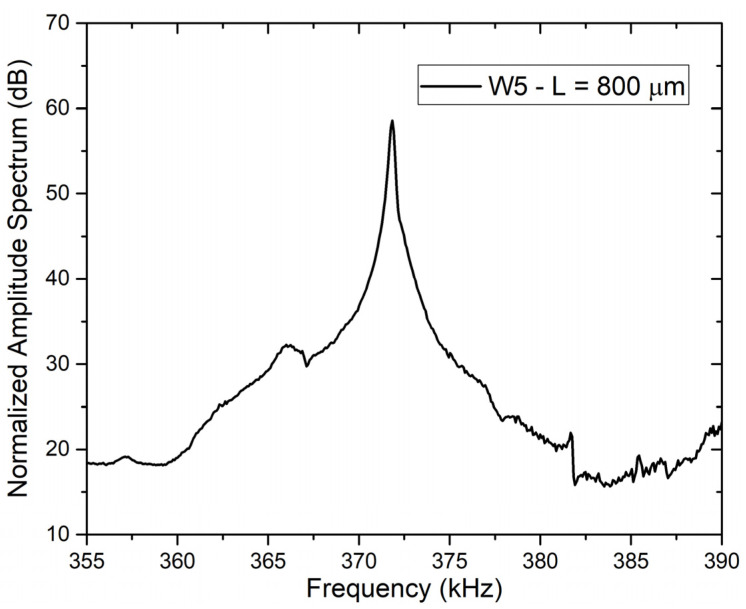
Normalized amplitude spectrum of 800 μm long double clamped beam around its resonance frequency.

**Figure 11 micromachines-12-01072-f011:**
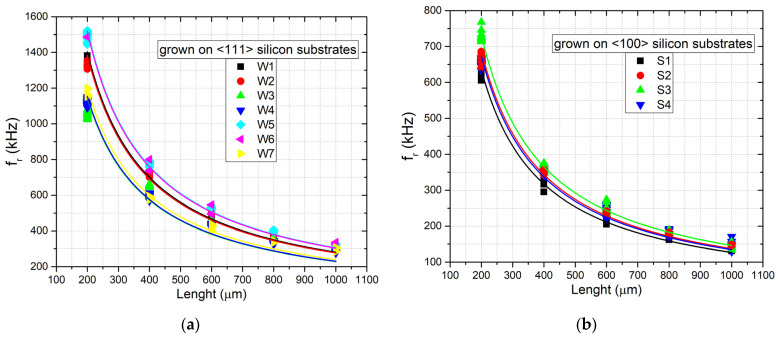
First-order resonance frequency versus length of the double clamped beams for the samples with n-type doping grown on (**a**) <111> and (**b**) <100> silicon substrates. The continuous lines are the data fittings obtained with Equation (2).

**Figure 12 micromachines-12-01072-f012:**
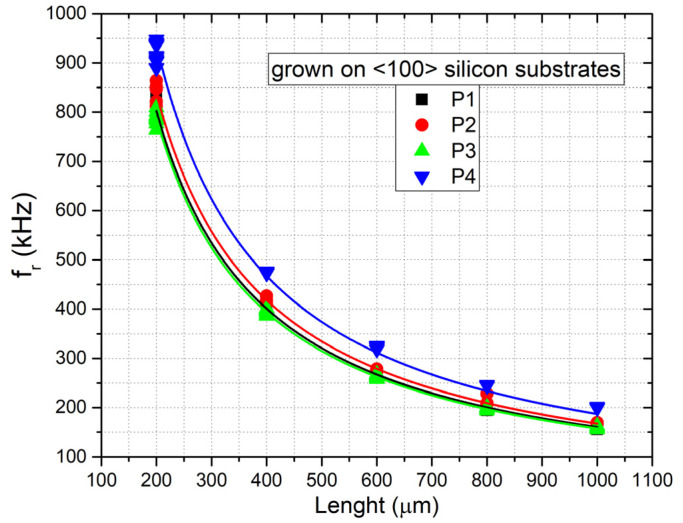
First-order resonance frequency versus length of the double clamped beams for the samples with p-type doping grown on <100> silicon substrates. The continuous lines are the data fittings obtained with Equation (2).

**Figure 13 micromachines-12-01072-f013:**
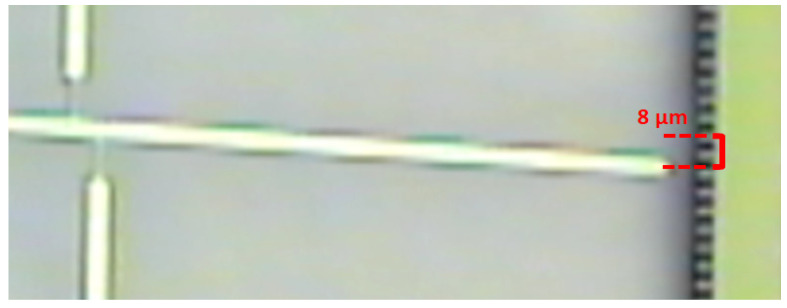
Shift of strain gauge indicating bar (sample P2) after release from the substrate.

**Figure 14 micromachines-12-01072-f014:**
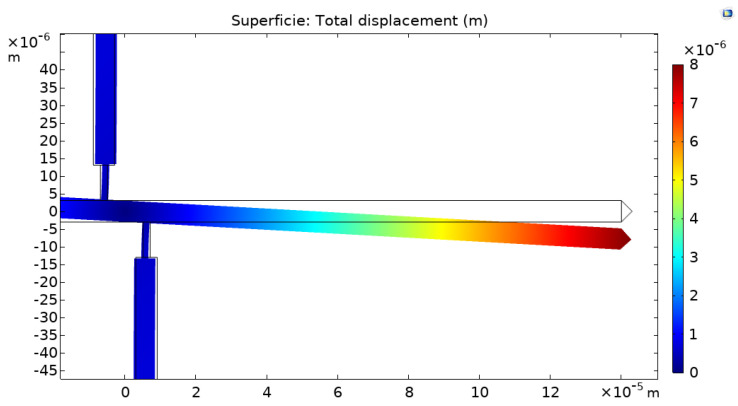
Simulated shift of strain gauge (P2 sample). In this case, Young’s modulus and the residual stress determined are 379 GPa and 360 MPa, respectively.

**Figure 15 micromachines-12-01072-f015:**
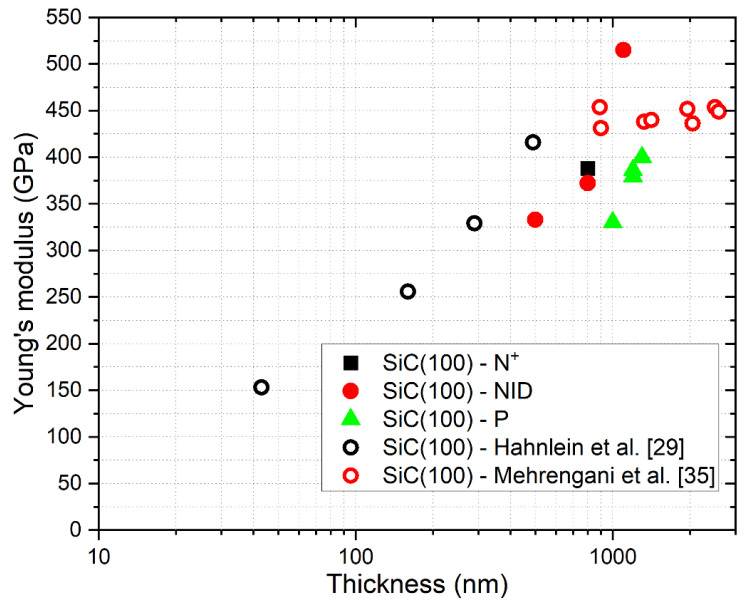
Comparison of measured values of E with other results from the literature about 3C-SiC grown on <100> silicon.

**Figure 16 micromachines-12-01072-f016:**
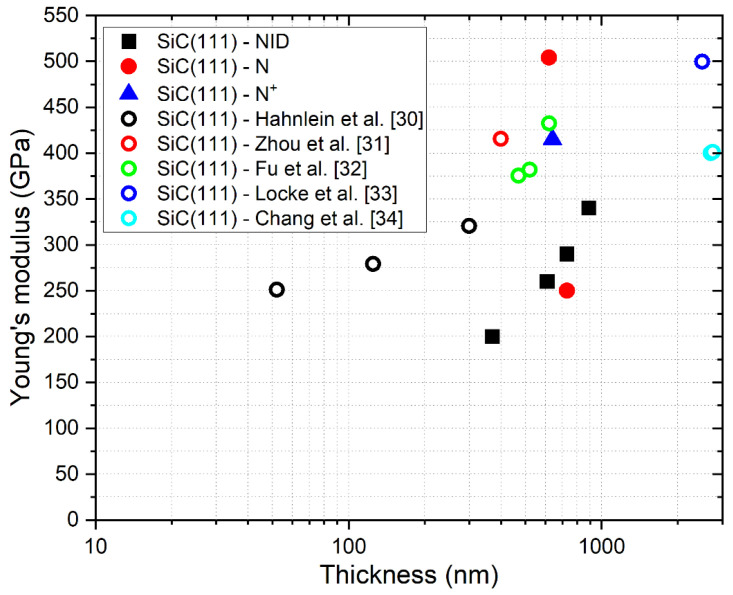
Comparison of measured values of E with other results from the literature about 3C-SiC grown on <111> silicon.

**Table 1 micromachines-12-01072-t001:** Overview of the 3C-SiC on Si samples prepared for the experiments.

Wafer Id	Substrate Type	Substrate Resistivity (Ω cm)	SOI Layer Resistivity (Ω cm)	Substrate Thickness	SOI Layer Thickness	SiC Layer Thickness	Sic Layer Doping
W1	<111> Si (bulk)	<0.01 (p)	-	1015 µm	-	0.89 μm	NID ^1^
W2	<111> Si (bulk)	<0.01 (p)	-	1009 µm	-	0.73 μm	NID ^1^
W3	<111> Si (bulk)	<0.01 (p)	-	1008 µm	-	0.37 μm	NID ^1^
W4	<111> Si (bulk)	<0.01 (p)	-	1008 µm	-	0.73 μm	N
W5	<111> Si (bulk)	0.001–0.01 (n)	-	1013 µm	-	0.62 μm	N
W6	<111> Si (bulk)	0.001–0.01 (n)	-	1016 µm	-	0.64 μm	N+
W7	<111> Si (bulk)	0.001–0.01 (p)	-	1017 µm	-	0.61 μm	NID ^1^
S1	<100> Si (SOI)	1–10(p)	>10,000 (i)	415 µm	10 µm	0.8 μm	N+
S2	<100> Si (SOI)	1–10 (p)	>10,000 (i)	415 µm	10 µm	0.8 μm	NID ^1^
S3	<100> Si (SOI)	1–10 (p)	>10,000 (i)	415 µm	10 µm	1.1 μm	NID ^1^
S4	<100> Si (SOI)	1–10 (p)	>10,000 (i)	415 µm	10 µm	0.5 μm	NID ^1^
P1	<100> Si (bulk)	1–10 (p)	-	517 µm	-	1.3 μm	Al
P2	<100> Si (bulk)	1–10 (p)	-	517 µm	-	1.2 μm	Al
P3	<100> Si (bulk)	1–10 (p)	-	518 µm	-	1.2 μm	Al
P4	<100> Si (bulk)	1–10 (p)	-	517 µm	-	1.0 μm	Al+

^1^ Not intentionally doped.

**Table 2 micromachines-12-01072-t002:** Geometrical parameters of the strain gauge.

Structure	Length	Width
Supporting bar	250 μm	6 μm
Pivot	10 μm	2 μm
Indicating bar	287 μm	6 μm

**Table 3 micromachines-12-01072-t003:** Overview of residual stress and Young’s modulus measurements on all the samples.

Wafer Id	Residual Stress	Young’s Modulus	Residual Strain
W1	1.01 GPa	340 GPa	$2.97\times 10^{-3}$
W2	982 MPa	290 GPa	$3.38\times 10^{-3}$
W3	674 MPa	200 GPa	$3.37\times 10^{-3}$
W4	682 MPa	250 GPa	$2.73\times 10^{-3}$
W5	1.2 GPa	504 GPa	$2.38\times 10^{-3}$
W6	1.18 GPa	415 GPa	$2.84\times 10^{-3}$
W7	738 MPa	260 GPa	$2.84\times 10^{-3}$
S1	207 MPa	388 GPa	$5.33\times 10^{-4}$
S2	242 MPa	372 GPa	$6.51\times 10^{-4}$
S3	274 MPa	515 GPa	$5.32\times 10^{-4}$
S4	233 MPa	333 GPa	$7.00\times 10^{-4}$
P1	331 MPa	400 GPa	$8.28\times 10^{-4}$
P2	360 MPa	379 GPa	$9.50\times 10^{-4}$
P3	319 MPa	386 GPa	$8.26\times 10^{-4}$
P4	450 MPa	330 GPa	$1.36\times 10^{-3}$

## Data Availability

The data reported in this article will be made available on the website of SiC Nano for PicoGeo project (grant agreement No 863220) http://picogeo.eu/.

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
