# Peer review of "Measurement of Residual Stress and Young’s Modulus on Micromachined Monocrystalline 3C-SiC Layers Grown on <111> and <100> Silicon"

_micromachines, 2021, doi:10.3390/mi12091072_

Round 1
Reviewer 1 Report
Paper: Measurement of residual stress and Young’s modulus on mi-2 cromachined monocrystalline 3C-SiC layers grown on <111> 3 and <100> silicon
Authors: Sergio Sapienza, Matteo Ferri, Luca Belsito, Diego Marini, Marcin Zielinski, Francesco La Via, and Alberto Roncaglia
Recommendation: Major revision - complete rewrite the article and explanation of obtained results.
To authors:
General comments:
The subject matter is of scientific interest. The article has two basic defects. The first defect is the complete chaos of the presentation. The second is the lack of support for conclusions through measurement results.
Detailed comments:
- The following sentence is inaccurate: ,,In fact, in the same paper it has been observed that this effect is 46 related to the improvement of the quality of the material or the decrease of the FWHM of the Raman TO peak.” What does it mean that FWHM of the Raman TO peak has changed? Let me refer directly to the structure of the material.
- The authors write in the Characterization section that XRD rocking curve measurements were performed for the selected samples. Unfortunately, no results or information (parameters) that have been obtained have been presented anywhere in the article.
- In general, the presentation of the measurement material in the article is very chaotic, and it needs to be corrected. The article is hard to read because it has no logical layout. In fact, it is not known how many samples have been prepared for measurements, a table with a list of samples is placed at the end of the article. These comments also apply to the description of the method of obtaining samples and the presentation of the results.
- Sections 2.2 and 3.2 have the same name: Fabrication. As I understand it, the first chapter deals with how to obtain test structures and the second chapter with the study of them. For example, in section 3.2 it was stated: '' The yield of the fabrication was generally good on all the wafers, even for the layers with higher residual stress. Moreover, the process flow described earlier allowed to obtain released structures with submicrometric size, both on gaps and beams, as was observed on the test DETF geometries designed on the mask (Fig. 8). Where did this information come from? How the measurement results were not presented? The following statements in this section are also not supported by the measurement results.
- The situation is similar with the Characterization sections: 2.3 and 3.3.
- Figures 12, 13 and 14 perfectly illustrate the chaos in the presentation. Namely, the result of the measurement of the resonance frequency of the double clamped beams has been presented, and nothing comes of it. It is not shown how this result translates into sample properties. And Figures 13 and 14 show the dependence on the length of the beam of resonant frequencies, from which no additional information can be derived. the fit also provides no new information except that the data fits well. In addition, Figures 13 and 14 give the impression that multiple independent systems have been tested, which is contrary to Table 5.
- One of the goals of the study was to determine the stresses. Nowhere in the presented results is there any direct information and determined values.
Reviewer 2 Report
The manuscript is showing the method to estimate Young’s modulus with independent measurements of hetero-epitaxial 3C-SiC grown on silicon film’s residual stress and strain that is reasonable. But some of data need to improve paper quality. Thus, this paper needs to improve before acceptance of this manuscript for publication in Micromachines.
- In this paper, the residual stress and Young’s modulus of 3C-SiC layers grown on <111> and <100> <100>silicon was reported to evaluate an achievable yield on the 3C-SiC layers in MEMS fabrication.But this design of experimental is complicated to reader. The doped 3C-SiC layers should deposit on the same orientation wafer (example <111>) to check the effect of doping types on stress and strain. Also the stress and strain depends on film thickness and substrate (thickness, insulation thickness of SOI and bulk resistance, pre-clean conditions). Therefore, I recommend that the authors should improve table 2 in page 15 for improving paper quality. And it should be discussed as well.
- 1 can remove for improving a paper quality and they are duplicated.
- Table 2 in page 6 and Table 2 in page 11 are duplicated (Table 2).
- “SOL” in line 76 (page 2) should change to “SOI.
- Please give reference no. in Fig. 17 and Fig. 18.
- The abbreviations and acronyms must be used at the first used word in the main text. Please up-date in main text (“TO”, “ FWHM”, etc).
Author Response
Please see the attachement

Round 2
Reviewer 2 Report
The manuscript is properly revised. The authors have responded positively to my observation and then, in my opinion, it can be accepted for publication. But many parameters in table 1 and 3 are duplicated, need to improve paper quality by combine in one table. Therefore, this paper needs to improve before acceptance of this manuscript for publication in Micromachines.
